# Safety and Immunogenicity of Combined DNA-Polyethylenimine and Oral Bacterial Idiotypic Vaccine for Patients with B-Cell Non-Hodgkin Lymphoma: A Pilot Study

**DOI:** 10.3390/cancers14143298

**Published:** 2022-07-06

**Authors:** Alexander Meleshko, Nadzeya Piatrouskaya, Katsiaryna Vashkevich, Dzmitry Lutskovich, Chuan Wang, Dmitri Dormeshkin, Natalia Savelyeva, Mikalai Katsin

**Affiliations:** 1Belarusian Research Center for Pediatric Oncology and Hematology, 223053 Minsk, Belarus; katsiaryna.vashkevich@gmail.com (K.V.); lutskovichdm@gmail.com (D.L.); 2Oncology Department of Czech Krumlov Hospital, 38101 Cesky Krumlov, Czech Republic; npetr1975@gmail.com; 3Department of Clinical and Molecular Cancer Sciences, ISMIB, University of Liverpool, Liverpool 11341, UK; chuan.wang@liverpool.ac.uk (C.W.); n.savelyeva@liverpool.ac.uk (N.S.); 4Institute of Bioorganic Chemistry of the National Academy of Sciences of Belarus, 220141 Minsk, Belarus; dormeshkin@gmail.com; 5Vitebsk Regional Cancer Centre, 210603 Vitebsk, Belarus; katin@mail.com

**Keywords:** idiotypic vaccine, polyethylenimine, lymphoma, DNA vaccine, *Salmonella*

## Abstract

**Simple Summary:**

Immunoglobulin variable domains, or idiotypes, have been used as lymphoma-specific antigens for therapeutic vaccination against B-cell lymphomas in a number of clinical trials. The effectiveness of DNA vaccines significantly depends on the chosen method of DNA delivery. In this study, we applied the intramuscular injection of a DNA–PEI vaccine followed by an oral vaccine-carrying *Salmonella* boost for lymphoma patients, which was safe and well tolerated. The observed remission was accompanied by T-cell but not an antibody response to the vaccine in most of the patients.

**Abstract:**

We report, in brief, the results of a phase I, non-randomized study of idiotypic DNA vaccination in patients with B-cell non-Hodgkin’s lymphoma (ISRCTN31090206). The DNA sequence of lymphoma-derived immunoglobulin variable regions was used as a tumor-specific antigen fused to the potato virus X coat protein. A conjugate of plasmid DNA with polyethylenimine was used for the intramuscular injections, followed by a boost with an oral live-attenuated *Salmonella* vaccine carrying the same plasmid. The patients with a complete or partial response to previous chemotherapy received one or two courses of vaccination, including four injections at monthly intervals. The vaccine was well tolerated, with low-grade adverse events. The T-cell immune responses were assessed by ELISpot, at last vaccine, one week and one month post-vaccination, and were detected in 11/14 (78.6%) of the patients. In cases of progression requiring chemotherapy, or the presence of a positive MRD after the first course of vaccination, the patients underwent a second course of vaccination. At the end point, 6/19 vaccinated patients had disease stabilization, while 13/19 were in complete remission. The overall survival was 100% at follow-up, of a median of 2.3 years.

## 1. Introduction

Despite recent advances in chemotherapy, indolent B-cell non-Hodgkin’s lymphomas (NHL) and chronic lymphocytic leukemia (CLL) remain incurable, with a slow progression and recurrent relapses, and a high percentage of patients with aggressive NHL still relapse. In 2015, the incidence of non-Hodgkin’s lymphoma in Belarus was 804 cases, including diffuse large B-cell lymphoma (301), small lymphocytic lymphoma (52), and follicular lymphoma (68). The three-year adjusted survival in 2019 was 53.3% for total NHL, 71.8 for follicular lymphoma, 53% for diffuse large B-cell lymphoma (DLBCL), and 42.3% for small lymphocytic lymphoma (SLL) [1].

A series of studies have shown the successful use of lymphoma-specific immunoglobulin (idiotype) as an antigen for therapeutic vaccination [2], including a successful phase III clinical trial [3]. While most of the clinical trials of idiotypic vaccines have used a protein preparation of the vaccine [4,5], DNA vaccines have also been tested in a number of clinical trials [6,7,8]. DNA vaccines have advantages, such as their ease of preparation, flexible design and safety, but their disadvantage is their low immunogenicity when injected alone. Synthetic carriers, such as cationic [9] and lipid [10] polymers, electroporation, needle-free injection [11], or delivery with a bacterial vehicle [12], have been used to enhance the immunogenicity of DNA vaccines.

In 2017, we announced the start of a phase I clinical trial of an idiotypic DNA vaccine for patients with B-cell non-Hodgkin’s lymphoma [13]. The trial used a DNA vaccine of a fusion gene encoding a single-chain variable fragment of the patient’s specific idiotype (Id) linked to the potato virus X coat protein (PVXCP) [14]. We also tested the combined administration of an intramuscular DNA-polyethylenimine (DNA–PEI) conjugate vaccine with an oral live-attenuated *Salmonella* vaccine to enhance the delivery of the DNA vaccine. Here, we report the safety, immunogenicity, and clinical outcomes after combined idiotypic DNA vaccination in patients with NHL and CLL.

## 2. Materials and Methods

### 2.1. Patients

Patient enrolment started in April 2017 and finished in September 2019. A total of 46 patients were assessed for eligibility. Twenty-four patients were not enrolled, according to the exclusion criteria previously described [13].

### 2.2. Clinical Protocol

The study was a phase I, non-randomized, open-label study of idiotypic DNA vaccination. The clinical protocol was approved by the National Cancer Center of Belarus and written informed consent was obtained from all patients. All patients had histologically confirmed diagnosis of B-cell non-Hodgkin’s lymphoma (MCL *n* = 12, DLBCL *n* = 8, FL *n* = 7, MZL *n* = 4) or CLL/SLL (*n* = 15). Lymph-node biopsy or bone-marrow aspiration (for CLL) were obtained at the initial diagnosis. Patients were treated according to the standard chemotherapy protocols accepted for the appropriate disease and stage. Patients with an established expression of a clonal immunoglobulin on tumor cells and having achieved a remission underwent vaccination 2–4 months after completing chemotherapy after the immune recovery.

The study objectives were to examine safety and tolerability, as well as to enhance the immunogenicity and therapeutic efficacy of the DNA vaccine through alternative methods of delivery of plasmid DNA.

### 2.3. Vaccine Production

Tumor tissues obtained by lymph-node or bone-marrow biopsies were used for the identification of lymphoma-specific idiotypes. Clonality and the isotype of tumor immunoglobulin was confirmed by flow cytometric immunophenotyping. A single-cell suspension containing approx. 250,000 cells was stained in 4 tubes: 1. Isotype control; 2. CD45-FITC, CD20-PE, CD3-PC5, CD19-PE-Cy7; 3. IgG-PE-Cy5, IgM-FITC, CD19-PECy7; and 4. kappa-FITC, lambda-PE, CD19-PE-Cy7 (BD Pharmingen, Franklin Lakes, NJ, USA). The immunoglobulin expression was estimated on lymphocytes as gated using SSC/FSC and CD19+. Variable region genes of heavy and light Ig were amplified as described previously [15]. Rearranged immunoglobulin gene segments were sequenced on ABI PRISM 3500 genetic analyzer (Thermo Fisher Scientific, Waltham, MA, USA) and identified using IMGT/V-QUEST v.3.5.30 and IgBLAST v.2.13.0 web tools. Sequences of variable regions (idiotypes) of patients’ Ig genes are given in the Appendix A. Either the PCR-amplified or synthetic sequence of variable domains were assembled as a linear fragment of single-chain variable fragment (scFv). Sequence encoding a secreted fusion of idiotype protein (scFv) and PVXCP was cloned into the pING vector. Vaccine construction incorporated the following components: a patient heavy chain leader peptide, VH-DH-JH fragment, linker L218 (GSTSGSGKPGSGEGSTKG) [16], light chain VL-DL-JL fragment, linker peptide AAAGPGP containing the NotI site [17], and the PVXCP sequence. The assembled plasmids were verified by restriction mapping and DNA sequencing. Extraction of plasmids was performed using MaxiPrep Kit (Thermo Fisher Scientific, Vilnius, Lithuania). Plasmid DNA was eluted using sterile DPBS (Thermo Fisher Scientific, Vilnius, Lithuania). Quality-control measures included agarose electrophoresis for plasmid isoforms (supercoiled form >80% DNA) and spectrophotometry (OD260/280 nm 1.8–2.0, OD260/230 nm 1.8–2.4) [18]. DNA concentrations were verified by Qubit Fluorometric Quantitation (Life Technologies, Waltham, MA, USA).

Linear PEI (20 kDa) (Sigma-Aldrich, Darmstadt, Germany) was used to prepare a complex with plasmid DNA at a *w*/*w* ratio of 1:1 (DNA:PEI), as described previously [13,19]. Plasmid DNA (500 μg) and PEI (50 μL of a 10-milligram/milliliter stock solution) were each diluted by 5% glucose to 4 mL. Complex formation was carried out by adding DNA solution to PEI dropwise before injections. The ratio 1:1 of PEI to DNA was calculated based on the mass of the polymer in the composition of the PEI hydrochloride used to prepare the solution, the molar ratio of N (PEI) to P (DNA) = 8.0. Charge and particle size measurements were performed on a Zetasizer Nano ZSP analyzer (Malvern Panalytical, Malvern, UK) equipped with Zetasizer Software v3.3, according to the manufacturer’s instructions. The transfection DNA–PEI particles had a size of 100–300 nm and a charge of +20–35 mV.

An oral form of vaccine was based on the newly designed (*aroA*, *guaAB*) attenuated strain of *Salmonella enterica* serovar typhimurium-SS2017 (genome sequence in GeneBank: CP053870–CP053874) electroporated with plasmid DNA [20]. The dose of 10^9^ CFU was measured for each vaccine preparation by serial dilution of overnight culture and seeding on plates for counting colonies. The bacterial suspension was washed with 0.9% saline, concentrated at 200 μL, resuspended in 50% glycerol, and loaded into gelatin capsules with cooling.

### 2.4. Minimal Residual Disease Monitoring

MRD detection has been described previously [21]. Clonal IgH and IgL gene rearrangements were identified during vaccine preparations. Allele-specific forward primers were selected to junctional regions and used in combination with a standard panel of germline primers and TaqMan probes (Primetech, Minsk, Belarus). RQ-PCR analysis of MRD was performed on Bio-Rad CFX96 (Bio-Rad, Hercules, CA, USA). DNA from follow-up samples of peripheral blood and bone marrow (if available) were amplified in triplicate for Ig target and albumin control genes. Interpretation of MRD results was performed according to European MRD Study Group guidelines [22]. MRD level was calculated in the follow-up blood samples relative to the target level in the diagnostic sample (tumor biopsy).

### 2.5. Immunological Evaluation

T-cell immune responses to the vaccine were measured by IFN-γ ELISpot assay. Cryopreserved pre- and post-vaccine PBMCs cells were thawed, stimulated with purified OKT-3 antibody (50 ng/mL, Exbio, Vestec, Czech Republic) for 3 days, and incubated in RPMI-1640 (Gibco, Vilnius, Lithuania) with 10% AB serum (Sigma-Aldrich, Darmstadt, Germany) in the presence of 300 U/mL recombinant IL-2 (Roncoleukin, LLC NPK BIOTECH, St. Petersburg, Russia) at 37 °C in 5% CO_2_ for one week. A 96-well plate (hydrophobic PVDF membrane, Merck Millipore, Darmstadt, Germany) was pre-wet with 70% ethanol, washed with sterile water 4 times, coated with 5 μg/mL primary antibody (mouse anti-human IFN-γ, BD Pharmingen, Franklin Lakes, NJ, USA) overnight at 4 °C, and blocked with complete medium containing 10% AB serum. Triplicates of PBMCs (1–4 × 10^5^) were stimulated with 20 μg/mL PVXCP protein, or 2 μg/mL PVXCP peptide pool (15-mer peptides with 11-mer overlap; Elabscience, Wuhan, China) or 10 μg/mL idiotype peptides in RPMI with 10% AB serum. For two patients, 11 Id peptides were synthetized as 15-mer peptides spanning the CDR3 regions of both immunoglobulin chains and protein regions containing non-germline amino acids (Appendix A). PBMCs (0.5–2 × 10^5^) were stimulated with 1 μg/mL CEFTA, CEF, and CMV peptide pools (Mabtech, Nacka Strand, Sweden) and 5 μg/mL PHA (Sigma-Aldrich, Darmstadt, Germany) as positive controls. Negative control was without stimulation (medium only). After a 48-hour incubation at 37 °C, plates were washed with 0.1% Tween 20 in PBS 4 times and incubated 1.5 h at 37 °C with 0.5 mg/mL secondary antibody (biotin mouse anti-human IFNγ antibodies, BD Pharmingen). IFNγ-secreting cells were detected with 1:1000-diluted streptavidin-alkaline phosphatase (BD Pharmingen) and BCIP/NBT substrate solution (Life Technologies, Waltham, MA, USA). Spot-forming cells (SFC)/well were counted using the AID EliSpot Reader iSpot Spectrum (AID GmbH, Strassberg, Germany). For each sample, the triplet mean number of spots per million cells was calculated, and the negative control was subtracted from the test value. A 3-fold increase in the number of IFNγ-producing T cells in the post-vaccine PBMC sample compared to the pre-vaccine sample was considered positive. Positive responses were defined as number of spots/million cells in the test ≥2 SDEV above the negative control.

Humoral responses were analyzed by ELISA. Plates coated with antigenic proteins (PVXCP or Id-scFv) were incubated with serial dilutions of patient serum and after washing were stained with HRP-labeled anti-human immunoglobulin antibodies (Thermo Fisher Scientific, Vilnius, Lithuania). A 3-fold increase in anti-vaccine antibody titer in post-immune serum compared with pre-immune serum was considered positive immune response.

## 3. Results

### 3.1. Patient Characteristics and Treatment Administration

Twenty-two patients meeting the inclusion criteria with established expression of clonal immunoglobulin on tumor cells were included in the study and underwent vaccination (Figure 1). Three patients were vaccinated with an aqueous DNA solution without a bacterial boost and were excluded from the analysis. Nineteen patients (12 males and 7 females) with various B-cell malignancies and a median age of 59 years (range 18–70 years) were enrolled in the vaccination protocol and received at least one vaccination course (Table 1). Four patients received two courses of vaccination. A total of 96 vaccinations were administered during the trial.

One vaccination course included four vaccine administrations at monthly intervals (Figure 2). At each vaccination, the patient received the DNA vaccine in two ways. First, a complex of 500 μg of plasmid DNA with linear PEI (DNA: PEI) was administrated by an intramuscular injection. The following day, a capsule containing 10^9^ CFU of an attenuated *Salmonella* strain carrying the same plasmid DNA, which acted as a carrier for enhancing intracellular delivery, was administered orally [24].

### 3.2. Safety and Adverse Events

The adverse events were assessed according to the Common Terminology Criteria for Adverse Events (CTCAE v5.0). The combination vaccine was safe and well tolerated by all the patients. Local pain reactions at the injection site lasting up to two days were observed in the majority of the patients. One patient had Grade 1–2 symptoms after vaccination: low-grade fever, headache, flu-like syndrome and nausea, symptoms of an inflammatory response likely caused by the DNA–PEI injection. One patient had an episode of Grade 1 diarrhea on the day of the vaccine, which resolved without treatment, possibly related to an oral bacterial vaccine. Overall, the oral administrations of the bacterial vaccine were asymptomatic and did not aggravate the symptoms of the intramuscular vaccine.

### 3.3. Immune Responses

Serum and PBMCs were collected from each patient at defined time-points for subsequent immunological analysis: pre-vaccination, on the day of last vaccination, one week, and 4–6 weeks after the last vaccination. The induction of T-cell-mediated immunity to the vaccine was evaluated; tumor-associated idiotype peptides were synthesized and assessed for two patients.

An ELISpot assay against PVXCP protein was performed for 14/19 patients. Eleven of the fourteen patients (78.6%) had a significant increase in spot number in at least one post-vaccination blood sample, including both of the patients for whom assessment with Id-peptides was possible (7T, 12O). The maximum immune response was observed one week (patient 14K, 12O) or one month (17K, 19V) after the last vaccine administration (Figure 3).

The antibody response was assessed by ELISA to PVXCP protein for 12 patients and recombinant scFv-PVXCP protein for two patients (3R, 4D). An antibody response to the PVXCP was detected in only two patients (2/12, 16.7%). The first patient (13S) was positive one month after the last vaccine administration. The increase in absorbance in the 50-fold serum dilution at this point was 5-fold compared to previous serum samples from this patient. The second (15B) was positive in all the time points and had significant pre-vaccine anti-PVXCP antibody titer (Figure 4).

The immune responses for all the patients are presented in the Appendix A.

### 3.4. Clinical Outcomes

The median follow-up was 27.8 (min 5.0–max 35) months from the last vaccination to the last follow-up. One patient (4D) started vaccination after the first relapse treated with R-CHOP + Benda. Eight months after the second remission, the vaccination course started. The patient was MRD-positive before the first course of vaccination and became negative after the second, but finally relapsed 615 days after the last vaccine. Another patient with MCL (3R) was MRD-positive after vaccination and progressed 35 days after the last vaccination of the first course. Subsequently, the patient received the second course and was observed for two years with stabilization of the disease and a constant level of MRD in the blood of about 10^−3^. The patients with CLL after treatment had measurable MRD levels. The CLL patients who did not achieve significant cytoreduction as a result of chemotherapy did not respond to the vaccine and subsequently progressed (6L, 11K) or remained stable (2K). At least three patients who were confirmed as positive for MRD before the start of the vaccination reduced their level of MRD after vaccination and remained in remission for an extended period (1F, 10S, 15B). According to our data, all the patients are alive. The follow-up failed to update in 2021 for three patients: 3R, 7T, and 9M.

## 4. Discussion

This is the first clinical trial of an idiotypic DNA vaccine administered as an intramuscular injection of a DNA–PEI conjugate followed by an oral *Salmonella* boost. The vaccines were composed of tumor-derived scFv linked to the virus gene, PVXCP, which was previously shown to enhance the immune response to Id [14,25]. Since plasmid DNA vaccines targeting cancer antigens have low immunogenicity and questionable efficacy when used as naked DNA, the enhancement of in vivo DNA vaccine delivery remains relevant. Combining plasmid DNA with a synthetic polymer to form a complex before administration to patients can facilitate the entry of DNA into cells, thus improving antigen expression, and represents an easily accessible and convenient alternative to the more commonly used method of electroporation [26,27]. In this study, we used a DNA conjugate with polyethylenimine 20 kDa for intramuscular administration to enhance the transfection efficacy of the DNA. Our preclinical studies have shown that DNA–PEI conjugates can enhance the immunogenicity of DNA vaccines in a mouse model [20,28]. The premise was to use a lower-molecular-weight 8-kilodalton PEI to produce a vaccine conjugate. However, since a 20-kilodalton PEI:DNA 1:1 *w*/*w* mixture was shown to be more stable and to provide the required size and charge of nanoparticles, we switched to a 20-kilodalton PEI, which yielded a higher transfection efficiency at a higher DNA/PEI ratio. Thus, it was possible to retain the desired dose of DNA (500 μg) but reduce the dose of the PEI, which would have been potentially toxic [29].

Our preclinical studies suggest that bacterial oral vaccine delivery can enhance the immunogenicity of DNA–PEI vaccines [20]. To further increase the vaccine immunogenicity, we added a booster vaccine consisting of an attenuated *Salmonella* enterica strain carrying the same plasmid as was used for the i.m. vaccination.

Several studies have demonstrated the ability of oral *Salmonella* vaccines to inhibit tumor growth in mouse models [24,30,31] and, further, to be safe and immunogenic, as was shown in at least one clinical trial, which used an anti-VEGFR-2 vaccine VXM01 in patients with pancreatic cancer [32,33].

Our data show that both intramuscular and oral vaccine delivery in this setting was well tolerated and caused minimal adverse symptoms.

Certain side effects were associated with the intramuscular injection of the DNA–PEI vaccine: systemic reactions, such as flu-like symptoms, headache, nausea, and fever occurred in only one patient after intramuscular injection and prior to the administration of the oral vaccine, and were likely related to the toxic effects of a large dose of polyethylenimine. Differing severities of pain or discomfort at the injection site were observed in most patients after injection, but were not accompanied by any other complications.

The administration of the oral vaccine was not associated with noticeable adverse effects, except for one incidence of diarrhea in a single patient (whose symptoms lasted for one day) after receiving the vaccine, which subsequently resolved without treatment. Thus, it can be concluded that the attenuated *Salmonella* strain used in the study is safe for vaccination.

The evaluation of the immunogenicity of the vaccine showed that our formulation induced detectable T-cell responses in the majority of the patients (78.6%), as assessed by ELISPpot. Only one patient developed an antibody response to the vaccine. We can assume that the lack of or weak antibody responses observed may be the result of the chosen DNA delivery methods (DNA–PEI, bacteria), which do not provide the large amount of soluble antigen protein needed to induce humoral immunity. Another reason is the administration of rituximab during chemotherapy prior to vaccination in most patients, which caused a long-term depletion of B-cells. In the absence of anti-Id antibody, the possible immune mechanism could be attributed to cytotoxic CD4 T cells, which would be in line with the pre-clinical data [14]. These are powerful effector T cells capable of eliminating cancer cells through a variety of mechanisms, including the direct killing of MHCII-positive lymphoma cells [34].

This study was a non-randomized, controlled trial. Therefore, it is not possible to draw an evidence-based conclusion about the therapeutic efficacy of the vaccines. However, the results obtained allow us to draw some preliminary conclusions. The combined DNA vaccine was shown to induce T-cell responses in most of the patients, and probably clinical responses, manifested in a decrease in MRD and in prolonged remission. A prolonged remission after vaccination was observed in three patients with mantle-cell lymphoma, which is usually associated with a poor prognosis; two of them remain in remission (3R—19.6 months, 17K—24.4 months), and the third (4D) relapsed 20.6 months after the last vaccination.

We also found that the vaccination of patients with CLL exhibiting a high number of leukemic cells in their blood, and without significant cytoreduction (one course of vaccination in patients 2K, 6L, 11K) has no therapeutic effect, presumably due to the high tumor load and immunosuppression caused by the tumor. Patients with CLL, following chemotherapy and after achieving remission, responded to vaccination and exhibited decreased MRD, which likely prolonged the remission period.

## 5. Conclusions

This study of patients with B-cell non-Hodgkin’s lymphomas and CLL demonstrated the safety and tolerability of a combined DNA–PEI vaccine and oral live-attenuated *Salmonella* DNA vaccine. The form of vaccination we used allowed us to induce T-cell immune responses in most of the patients, but failed to induce measurable antibody responses. The good clinical results indicate that T-cell immune responses may be sufficient to consolidate remission.

## Figures and Tables

**Figure 1 cancers-14-03298-f001:**
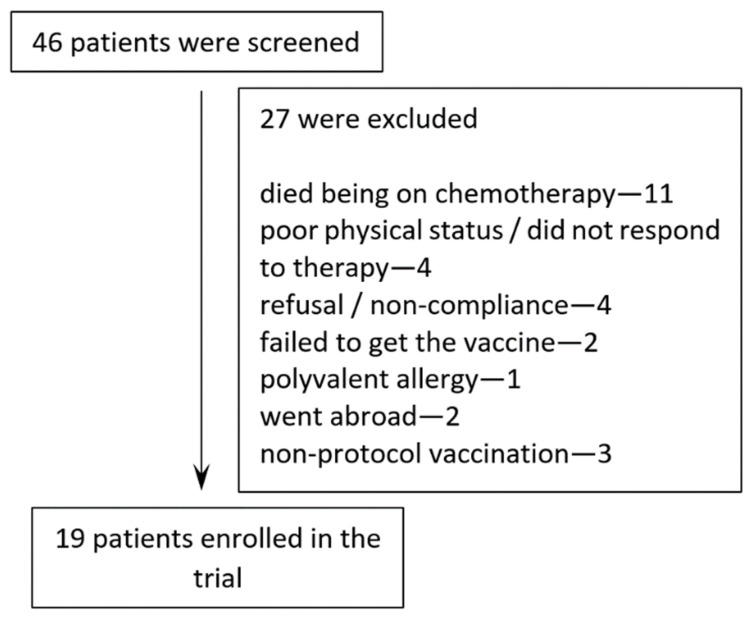
Patient cohort included in the study.

**Figure 2 cancers-14-03298-f002:**
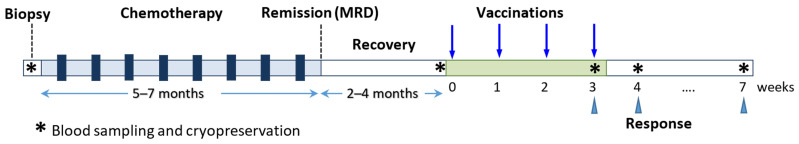
Overall treatment scheme. MRD—minimal residual disease.

**Figure 3 cancers-14-03298-f003:**
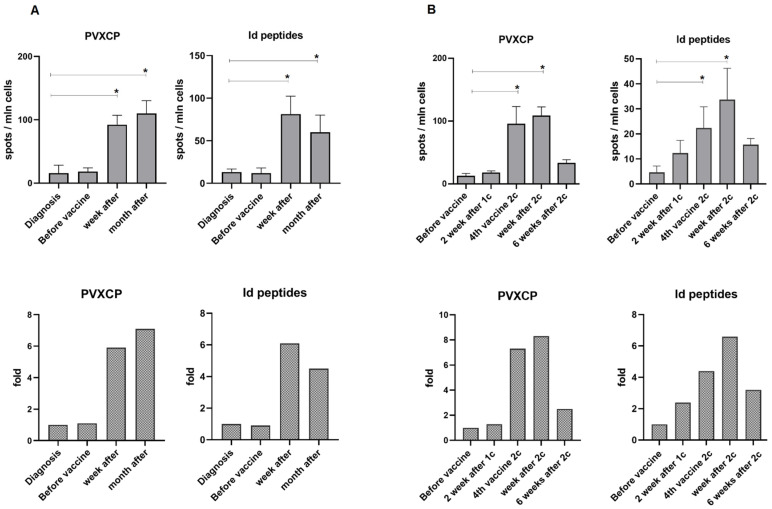
ELISpot test for patients 12Os (**A**) and 7T (**B**). The T-cell response is presented as the number of spots per million cells at each point (mean and SD), and the mean fold increase in response relative to the primary point. There are no data for the fourth vaccine of the patient 12Os. 1c—first vaccine course, 2c—second vaccine course (indicated only for patients who had two courses of vaccination). A significant difference (*p* < 0.05) in the number of spots between time points is indicated by the asterisk (Student’s *t*-test).

**Figure 4 cancers-14-03298-f004:**
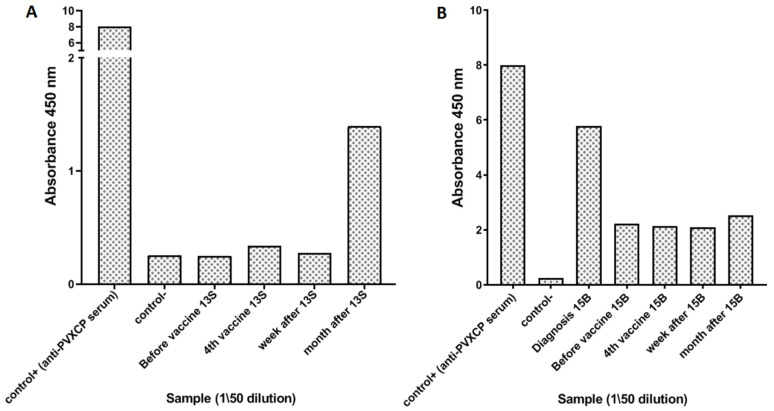
Anti-Id antibody responses for patients 13S (**A**) and 15B (**B**).

**Table 1 cancers-14-03298-t001:** Results of treatment and vaccination of the patients.

Patient	Age ^1^	Diagnosis	Tumor Ig	Chemotherapy	Response to Chemo	Time from Last Cycle of Chemo (Month)	VC	Adverse Events	Immune-Response ELISpot (Max Vac/Prevac Fold)/ELISA	MRD Status Post-Chemo	MRD after 1 Course	MRD after 2 Courses	Post-Vaccine Status	Last Follow-Up	Time from Vaccination (Months)
1F	M/57	CLL	IgM,D/IgK	FCR	PR	5	1	Local Grade 1	negative/n.d.	0.76%	0.05%		stabilization	remission	34
2K	M/60	CLL	IgD/IgK	FCR	PR	10	2	Local Grade 1	n.d.	249%	563%	n.d.	stabilization	remission	33
3R	M/52	MCL	IgM,D/IgK	EPOCH_BAC	CR	12	2	Flu-like syndrome, nausea (grade 2)	positive (10)/negative	1.7%	0.57%	0.2%	stabilization	n.d.	20
4D	M/67	MCL	IgM,D/IgK	BCDP + R-CHOP + Benda	CR	9	2	Local Grade 1	positive (5.4)/negative	0.05%	0	0	remission	relapse 2	33
5S	F/43	SLL	IgG/IgL	FCR	CR	1	1	Local Grade 1	positive (7.6)/n.d.	n.d.	0		remission	remission	22
6L	M/63	CLL	IgM,D/IgK	FCR	PR	4	1	Local Grade 1	n.d.	34%	275%		stabilization	progression	28
7T	F/50	MZL	IgM/IgKlow	FCR	CR	5	2	Diarrhea (1)	positive (8)/negative	0	0	0	remission	n.d.	8
8S	F/58	SLL	IgM/IgK	FCR	CR	9	1	Local Grade 1	n.d./negative	n.d.	n.d.		remission	remission	32
9M	M/65	SLL	IgM/IgL	FC	CR	11	1	Local Grade 1	negative/negative	0	0		remission	n.d.	5
10S	M/70	CLL	IgM,D/IgK	Benda + Ganziva	CR	5	1	Local Grade 1	uncertain (2.4)/negative	0.0175%	0		remission	remission	33
11K	F/63	CLL/SLL	IgD,M/IgL	FCR	SD	10	1	Local Grade 1	n.d./negative	16%	15%		stabilization	progression	16
12O	M/64	FL	IgM/IgK	BR	CR	6	1	Local Grade 1	positive (6)/negative	n.d.	0		remission	remission	31
13S	M/18	DLBCL	IgG/IgK	B-NHL-M [23]	CR	3	1	Local Grade 1	positive (3.6)/positive	0	0		remission	remission	32
14K	F/35	FL	IgM/IgK	BR + R-CHOP	CR	4	1	Local Grade 1	positive (14.3)/negative	0	0		remission	remission	5
15B	M/59	FL	IgM/IgK	BR	CR	7	1	Local Grade 1	positive (16.9)/positive	0.024%	0		remission	remission	28
16B	M/69	DLBCL	IgM/IgK	R-CHOP + radiotherapy	CR	6	1	Local Grade 1	n.d./negative	0	0		remission	remission	35
17K	M/62	MCL	IgM/IgK	BR plus rituximab	CR	6	1	Local Grade 1	positive (4.9)/negative	n.d.	0		remission	remission	24
18B	F/39	DLBCL	IgM/IgL	R-CHOP	CR	5	1	Local Grade 1	positive (10.2)/negative	0	0		remission	remission	24
19V	F/18	DLBCL	IgG/IgL	B-NHL-M [23]	CR	6	1	Local Grade 1	positive (9.4)/negative	0	0		remission	remission	12

^1^—age at the start of vaccination, VC—vaccine courses, DLBCL—diffuse large B-cell lymphoma, SLL—small lymphocytic lymphoma, FL—follicular lymphoma, MZL—nodal marginal zone B-cell lymphoma, MCL—mantle cell lymphoma, PR—partial remission, CR—complete remission, SD—stable disease, n.d.—no data.

## Data Availability

The data presented in this study is available within the article or Appendix A.

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
