# Peer review of "Safety and Immunogenicity of Combined DNA-Polyethylenimine and Oral Bacterial Idiotypic Vaccine for Patients with B-Cell Non-Hodgkin Lymphoma: A Pilot Study"

_cancers, 2022, doi:10.3390/cancers14143298_

Round 1
Reviewer 1 Report
In this study, Meleshko et al report the results of a phase I, non-randomized clinical trial of id-pxv fusion DNA vaccines in patients with B-cell lymphoma and CLL. They report that the vaccine was well tolerated with adverse events of modest clinical significance. ELISpot T-cell immune responses were detected in 13/15 patients and 6/19 vaccinated patients had disease stabilization following vaccination. Overall survival was 100% but follow-up was very short (median 2.3 years).
The therapeutic approach is very interesting and deserves publication, possibly pending revision of the following points:
1) the follow-up is too short and the population too small to drive any conclusion on overall survival
2) the authors mention MRD. Was MRD measured systematically pre- and post-vaccination in all patients? this should be shown in table 1 if available.
3) some sections are hard to read: e.g. section 3.1: "This section may be divided by subheadings. It should provide a concise and precise description of the experimental results, their interpretation, as well as the experimental conclusions that can be drawn." needs to be removed. This section may be divided by subheadings. It should provide a concise and precise description of the experimental results, their interpretation, as well as the experimental conclusions that can be drawn." needs to be removed.
4) the authors claim that ELISpot T-cell immune responses were detected in 13/15 patients. However it is not clear in how many patients anti-Id specific immune responses were measured and in how many a response was detected.
5) Similar problem as per point 4 for antibody responses. also it is not clear if there were patients who had no measurable pre-vax antibody titers and mounted an immune response to PVXCP following vaccination.
6) the manuscript has several grammatical errors and may merit professional revision of the English?
Reviewer 2 Report
In this manuscript authors report results of phase I clinical trial testing DNA vaccine encoding patient specific lymphoma clone Idiotype (Id) as a soluble single-chain variable fragment fused to the potato virus X coat protein (PVXCP) for Non Hodgkin B-cell Lymphoma. First dose of the vaccine was administered intramuscularly in the form of polyethylenimine (PEI) complexes followed by oral administration of second dose of vaccine in the form of live attenuated Salmonella vaccine carrying plasmid DNA. Each course of treatment consisted of 4 rounds of intramuscular and oral vaccinations at monthly intervals. Study objectives were to examine safety and tolerability of vaccination and to enhance immunogenicity and efficacy. A total of 19 patients were enrolled for final vaccination protocol. Safety of the vaccine was determined based on the occurrence of adverse events. Authors indicate that vaccine was generally well tolerated with most patients experiencing Grade 1-2 symptoms. Authors analyzed T and B cell immune responses to vaccine were using blood samples pre- and at different time points post-vaccine. T cell responses were analyzed with ELISpot on PBMC samples using either PVXCP peptides or patient Id peptides for two patients. Authors indicate that 11 out of 19 patients tested (78%) showed T cell responses to PVXCP peptides and two out of two tested patients showed T cell responses to Id peptides stimulation. Maximum response was observed at one week or one month time point following the end of vaccination regiment. PVXCP antibody response was observed in 2 out of 12 patients tested including one patient with pre-existing anti-PVXCP antibodies. Overall, authors indicate that tested vaccination regiment is safe and well tolerated and induces T cell responses. This is a study that potentially introduces a new vaccination strategy of DNA-based anti-idiotype vaccine to improve feasibility of current approaches.
Authors should address few major comments:
1. The study design suffers from the limited sample size (only 2 of 19 patients) available for analysis of antigen-specific responses against the tumor antigen of interest (idiotype). Assuming the samples are available, additional patients should be added to the analysis.
At a minimum, Authors should include data panel of ELISpot of T cell response to patient-specific Idiotype peptides for the second patient that was evaluated for anti-Idiotype responses which is mentioned in the text to further prove that this vaccination regiment generates T cell responses against idiotype antigen.
The claim of idiotype-specific responses should be supported by lack of reactivity to a panel of unrelated idiotype peptides
2. Authors should include data panel of ELISpot of T cell responses to PVXCP for other patients that were evaluated (either in main manuscript or supplementary data) to further confirm that vaccination is effective at inducing T cell responses.
3. Results of antibody response analysis should be shown for patient that showed positive response in main figure (mentioned in text). Additionally, data from negative patients could be shown in supplementary materials.
Minor comments:
1. The text of the manuscript could be rearranged a little so that some of the information from Materials and Methods that contains references to figures could be moved to results section. For example, figure 2 along with parts of section “Clinical protocol” of materials and methods that describe figure 2 could be moved to Results section “Patient characteristics and treatment administration”.
2. Line 93 should be “lymph node or bone marrow” instead of “of”.
In the introduction, reference to the positive Phase 3 randomized clinical trial of idiotype protein vaccination should be added (Schuster et al. J. Clin Oncol 2011)
Reviewer 3 Report
The article reported the results of a phase I clinical trial of an idiotypic DNA vaccine for patients with B-cell lymphoma and CLL. The novelty of the vaccine design includes a conjugate DNA-PEI, linkage to immunostimulatory sequence PVXCP (conjugate too?), and oral live attenuated salmonella vaccine (adjuvant). The results showed promising safety, immunologic responses, and potential positive effect on clinical outcome. The article is well written and I only have a few comments below:
(1) Approximately 40% CLL patients have unmutated IGHV, which could be relevant for the low vaccine efficacy in the entity. How was the mutation status of the CLL patients analyzed in the clinical trial?
(2) Figure 2 did not show the time range of the scheme.
(3) For vaccine production methods session, how clonality was analyzed? Did tumors have only one tumor clone (isotype) for one patient? When there were multiple tumor clones or derivative of the dominant clone’s isotype, how were the vaccine sequences determined?
(4) For the immunological evaluation, immune response to all conjugates and adjuvant should be tested, especially the humoral response to Salmonella.
Only two patients were assessed for Id-specific immune response (line 200 in page 5)?
In the discussion, the author mentioned the cytotoxic CD4 T cells: could this be tested?
(5) Abbreviations underneath Table 1 were not complete, especially, chemotherapies (What is B-NHL-M regimen?) and VC2 were not mentioned in the manuscripts but were not included in the notes.
Round 2
Reviewer 2 Report
Authors promptly addresses reviewer’s concerns by including additional figure panels in main
manuscript and supplementary data and including additional reference.
Additional minor comments:
1. In the abstract (line 27) authors state that vaccination regiment consisted of 3
injections, whereas the following result section (line 173) indicated 4 and so is figure 2.
Please adjust to make it consistent.
2. In result section lines 193-194 authors could edit or remove “pre-vaccination, on the day
of last vaccination, one week and four weeks post last vaccination” because in figure 3
there are other time points shown (for example 6 weeks post) which makes it confusing.
3. In figure legend to Figure 3 authors should include what ‘1c’ and ‘2c’ mean, similar to
how it is explained in supplementary figure S1.
4. In the discussion section (line 277) authors mention 87% positive T cell response but
elsewhere it is 78% (abstract line 29, results line 198). Please adjust to make consistent.
